# *Helicobacter pylori* in Native Americans in Northern Arizona [note 1]

**DOI:** 10.3390/diseases10020019

**Published:** 2022-03-23

**Authors:** Fernando P. Monroy, Heidi E. Brown, Priscilla R. Sanderson, Gregory Jarrin, Mimi Mbegbu, Shari Kyman, Robin B. Harris

**Affiliations:** 1Department of Biological Sciences, College of the Environment, Forestry and Natural Sciences, Northern Arizona University, 617 South Beaver Street, Flagstaff, AZ 86011, USA; priscilla.sanderson@nau.edu; 2Pathogen and Microbiome Institute, Northern Arizona University, Flagstaff, AZ 86011, USA; mimi.mbegbu@nau.edu (M.M.); shari.kyman@nau.edu (S.K.); 3Department of Epidemiology and Biostatistics, Mel and Enid Zuckerman College of Public Health, 1295 N Martin Ave, Tucson, AZ 85724, USA; heidibrown@arizona.edu (H.E.B.); rbharris@arizona.edu (R.B.H.); 4Winslow Indian Health Care Center, 500 North Indiana Avenue, Winslow, AZ 86047, USA; tubasurgeon@yahoo.com

**Keywords:** *Helicobacter pylori*, gastric cancer, ulcers, gastritis, Navajos, Native Americans, American Indians, virulence factors

## Abstract

Background: In Arizona *Helicobacter pylori* prevalence of infection among Navajo adults is about 62% and gastric cancer incidence rate is 3–4 times higher than that of the non-Hispanic White population. Aim: The aim of this study was to estimate the prevalence of specific *H. pylori* virulence factors (*cagA* and *vacA*) among Navajo patients undergoing and their association with gastric disease. Methods: Virulence genes, *cagA* and *vacA,* in *H. pylori* were investigated in gastric biopsies from 96 Navajo patients over age 18 who were undergoing esophagogastroduodenoscopy. Biopsies from the antrum and fundus were used for molecular characterization to determine *cagA* type and number of EPIYA motifs and presence of alleles in the signal (s) and medium (m) regions of the *vacA* gene. Results: *H. pylori* infection was found in 22.9% of the biopsy samples. The *cagA* gene amplified in 57.6% of samples and showed a predominant “Western *cagA*” type, with the EPIYA-ABC motif (45.4%), most prevalent. The *vacA* allele s1bm1 was the most prevalent (54.5%). Conclusions: *H. pylori* genotypes were predominantly *cagA* Western-type and ABC EPIYA motifs. The *vacA* s1bm1 genotype was the most prevalent and seemed to be associated with gastritis. American Indian/Alaska Native populations are at higher risk for gastric cancer. It is important to identify genotypes of *H. pylori* and virulence factors involved in the high prevalence of *H. pylori* and associated disease among the Navajo population.

## 1. Introduction

*Helicobacter pylori (H. pylori)* is a fastidious microaerophilic Gram-negative rod. Infection with *H. pylori* is among the most common bacterial infections with nearly half of the global population estimated to be infected [1,2]. Colonization by the bacterium primarily occurs during childhood; an estimated 50% of children in developing countries and 10% of children in the United States have been colonized before the age of 10 [3,4]. Prevalence of infections with *H. pylori* is highly variable owing to age, race/ethnicity, gender, and socioeconomic factors, with most suggesting the higher prevalence in older age groups results from living conditions during their childhood [5,6].

Transmission of *H. pylori* has been reported to occur via gastro-oral, oral-oral, and fecal-oral routes [7,8]. Colonization of gastric mucosa is asymptomatic in most individuals and can persist for several decades [9,10]. Infections can cause mild to severe mucosal inflammation and are a risk factor in the etiology of several severe gastrointestinal outcomes including, peptic ulcer disease, chronic gastritis, and stomach or gastric cancer [1]. While a minority of chronically infected individuals develop gastric cancer [11], *H. pylori* infection has been attributed to nearly 89% of all non-cardia gastric cancers which represents over three-quarters of all gastric cancer worldwide [12]. Development of malignancy is a result of a complex interaction between environmental factors, host genetics, and virulence of the infecting *H. pylori* strain resulting in enhance or reduced inflammatory responses [13,14].

Colonization of gastric mucosa is accomplished by the action of virulence factors, (outer membrane proteins and adhesins) which enable adherence to the gastric epithelial cells [15,16]. Damage to the epithelium comes primarily from two well-studied virulence factors, the cytotoxin-associate gene A (*cagA*) and the vacuolating cytotoxin A gene (*vacA*) [17,18]. The *cagA* protein is one of the products encoded by the cag pathogenicity island (cag-PAI) which is responsible for the translocation of *cagA* to the cytoplasm of gastric epithelial cells in *cagA*-positive strains of *H. pylori* [19,20]. The severity and prevalence of gastric diseases are associated with *cagA*-positive *H. pylori* strains [21,22,23]. The prevalence of *cagA*-positive strains varies between ethnic groups and regions; in Asia *cagA*-positive strains reach >90% compared to 50–60% in Western countries [24,25,26]. The pro-inflammatory and carcinogenic activities of *cagA* are associated with the number and type of phosphorylation sites denominated EPIYA (Glu-Pro-Ile-Tyr-Ala) regions [18,27,28]. According to the amino acid sequences that flank this region, EPIYAs are classified as EPIYA-A, -B, -C, or -D. Western *H. pylori* strains preferentially express *cagA* containing EPIYA-A, and -B with one or more -C segments; while strains from Asia contain EPIYA-A, -B, and D motifs [27,29]. Gastric pathology is associated with EPIYA domains containing more than one EPIYA-C phosphorylation site [30,31].

The *vacA* protein is encoded by the *vacA* gene, present in all strains of *H. pylori* and variability in the 5′ signal (s) and middle (m) regions results in alleles that when expressed together increase the risk for gastric pathologies [17,32]. The combination of *vacA* alleles s1m1 results in strains producing high levels of this virulence factor; the s1m2 strains produce moderate levels, while the s2/m2 strains produce minimal concentrations or do not produce it at all [33,34]. The s1m1 and s1m2 genotypes generate *vacA* isoforms that cause direct damage to the gastric epithelium and stimulate an acute inflammatory process, which may lead to chronic gastritis or gastric ulcer [35,36]. The prevalence of genotypes of *H. pylori* that express virulent *vacA* isoforms varies with the geographic area, and infection with *H. pylori vacA* s1m1 type correlates with increased risk of gastric disease [26,37].

Despite the elevated burden of gastric cancer in certain communities and populations and the established link between *H. pylori* infection and no-cardia gastric cancer, data on *H. pylori* prevalence in Indigenous communities in the United States are sparse [26,38,39]. The Navajo Nation is a sovereign Native American nation with the largest territory in the United States. Prevalence of *H. pylori* in Navajo Nation has been reported at 68–70% [40] which is close to values found in Alaska Natives, 75% [38,41,42]. Perhaps what is more striking about the high *H. pylori* infection prevalence is that gastric cancer incidence is 3–4 times higher in Navajo Nation than among the non-Hispanic white population in Arizona [43]. These high rates of chronic infection likely contribute to disproportionately high rates of gastric cancer reported in Navajos and Alaska Natives [44,45,46,47].

Despite the high prevalence of *H. pylori* in these populations, there are few reports on the association between infection and gastroduodenal diseases, and still fewer on the types and distribution of *vacA* and *cagA* genotypes and clinical outcomes in patients with gastritis, peptic ulcers, or gastric cancer [26,48,49]. The aim of this study is to determine the genotypes of *H. pylori* present in Navajo adults in Northern Arizona and evaluate their association with clinical outcomes. Identifying the distribution of genotypes of *H. pylori* in Navajos will facilitate therapeutic and prevention strategies to improve clinical outcomes.

## 2. Materials and Methods

### 2.1. Ethical Considerations

We obtained resolutions of support and approval prior to community recruitment from the three participating Navajo chapter communities (chapters) and the two governing agency councils that incorporated the chapters. These resolutions were required before receiving the Navajo Nation Human Research Review Board for protocol approval. The Northern Arizona University Institutional Review Board also approved the final protocol and consent forms.

### 2.2. Patients and Sample Collection

Participants were recruited when visiting the gastroenterology clinic at Winslow Indian Health Care Clinic (WIHCC) as part of their already scheduled pre-procedure visit for esophagogastroduodenoscopy (EGD). The study was explained to the participants by the clinic nurse and consent forms signed prior to the EGD. Consent was received to obtain additional biopsy samples and to review medical records for histopathology findings, prior testing for *H. pylori*, and demographic characteristics. All patients signed an Informed Consent Form. The only exclusion criteria were that participants must be at least 18 years of age. Clinic staff completed medical record abstraction and removed all personal identifying information using a standardized data form. All data and samples were deidentified prior to transmission to the university laboratories for analysis and data entry.

During endoscopy, four biopsies were collected, two from the antrum and two from the fundus. One set of each was submitted for histopathological examination to be performed by an established laboratory contracted by WIHCC. The other set was placed in 400 µL RNAlater for transport to the NAU laboratory and DNA isolation.

### 2.3. H. pylori Strains and DNA Isolation

The strains *H. pylori* 26695 (ATCC 700392) and *H. pylori* 60190 (ATCC 49503), were used as reference isolates. Both strains have the genotype *vacA* s1/m1 and are *cagA*+. Biopsies were collected from the antrum and the fundus regions in separate tubes containing RNAlater (Sigma. Chemical Co., St Louis, MO, USA). DNA was isolated using the FastDNA Spin Kit (MP Biomedicals, LLC, Solon, OH, USA) and the FastPrep 24 instruments (MP Biomedicals LLC) as described by the manufacturer. DNA concentration was measured in a Nanodrop 1 (Thermo Fisher Scientific, Waltham, MA, USA) and stored at −20 °C until used.

### 2.4. Molecular Identification

The DNA samples were used to first identity the presence of *H. pylori* by PCR amplification of the *H. pylori* 16S rRNA gene. Positive and negative controls were included in each reaction. In some cases, the ~519 bp-amplified product was separated by electrophoresis in 1% agarose gels followed by SYBR Green staining and analysis under an ultraviolet (UV) light. Genotyping of positive strains for the identification of *cagA* and *vacA* genes was carried out as previously described [33,50] using the primers shown in Table 1.

### 2.5. CagA Gene Amplification

All *H. pylori* 16S rRNA gene-positive samples were used in PCR to detect the *cagA* and *vacA* genes using primers described in Table 1. Amplification to detect presence of the *cagA* was achieved by targeting the constant region using primers F1 and B1 [50,56]. A 349-bp product indicated presence of *cagA*. To determine the variability in the EPIYA domains, the 3′ variable region was amplified using primers cag2 and cag4 resulting in products of 550-850-bp [49,50,51,52,53,54,55,56]. The PCR reaction mix consisted of 1.7 mM MgCl_2_, 0.2 mM dNTPs, 1 U of Platinum^®^ Taq DNA polymerase (Invitrogen, Carlsbad, CA, USA), and 300 ng of total DNA in a total volume of 25 μL. The PCR conditions used for amplification were: 1 cycle at 94 °C for 5 min; 35 cycles at 94 °C for 40 s, 56 °C for 30 s, and 72 °C for 50 s; and a final extension cycle at 72 °C for 7 min. The PCR products were subjected to electrophoresis on a 1.5% agarose gel and analyzed under an ultraviolet (UV) light. Samples were considered *cagA*-positive when at least 1 of the 2 bands was observed.

### 2.6. vacA Gene Amplification

PCR analysis was performed on *H. pylori* DNA samples to genotype the *vacA* s and m regions and to detect the presence of the *cagA* gene using previously described primers (Table 1) [33,50]. Real-time PCR primers were tested for specificity against DNA collected from *Helicobacter*, *Burkholderia*, and *Campylobacter*. Genotyping of *vacA* was assessed by PCR with oligonucleotides specific for each region (Table 1).

The reaction mixture contained 1.5 mM MgCl_2_; 0.2 mM dNTPs; 5 pmol of oligonucleotides VAGF and VAGR, or 2.5 pmol of VAIF and VAIR, 1.5 U of Taq DNA polymerase and 200 ng of DNA, in a total volume of 25 μL. Amplification conditions were: 1 cycle at 94 °C for 5 min; 35 cycles at 94 °C for 1 min, 57^o^ C for 1 min, 72 °C for 1 min; and a final extension cycle at 72 °C for 10 min. The PCR products were subjected to 1.5% agarose gel electrophoresis, stained with SYBR Green and visualized with ultraviolet light using a transluminator. In each PCR, DNA from strain 60190 (*vacA*s1m1/*cagA*^+^) was used as a positive control, and as a negative control, DNA was replaced with sterile deionized water. All reactions were performed in a gradient iCycler (BioRad, Hercules, CA, USA). The reference strains 26695 (GenBank AE000511), 60190 (GenBank U05676.1) and Tx30a (ATCC 51932) were used in the phylogenetic analysis of nucleotide and amino acid sequences, these strains are available in the NCBI (https://www.ncbi.nlm.nih.gov/ accessed on 15 June 2021). Amplified PCR products were gel-purified and subjected to sequencing with both forward and reverse primers using BigDye technology on an AB13700XL DNA sequencer (GeneWis, South Plainfield, NJ, USA).

### 2.7. Statistical Analysis

For all statistical comparisons, persons infected with mixed allelic subtypes (i.e., s1a/s1b) were categorized as s1b. Also, persons infected with both *cagA*-positive and *cagA*-negative *H. pylori* were categorized as infected with a *cagA*-positive strain. For the correlation of *H. pylori* genotypes with clinical data, we restricted the evaluation to three combinations of *vacA*s and *vacA*m regions with adequate sample sizes for evaluation: s1m1, s1m2, and s2m2. We compared clinical variables between these three genotypes with the likelihood ratio x^2^ test and analysis of variance (ANOVA) for categorical and continuous variables, respectively. All *p* values were two sided, and a *p* value of < 0.05 was considered statistically significant.

## 3. Results

Ninety-six (96) gastric biopsies sets were collected at WIHCC during EGD procedures during a 11-month timeframe (01/2018–05/2019; Table 2).

There were no differences observed in mean age between those with positive biopsy samples for *H. pylori* (54 ± 12.4) or negative (56.1 ± 12.5). Women were less likely to have a positive biopsy (*n* = 12/71) compared with men (*n* = 10/25; *p* = 0.22). The frequency of *H. pylori* isolation was 22.9%. Most patients came from Leupp (26.4%), followed by Winslow (18.8%), with no difference by biopsy outcome (Χ^2^ = 9.9, *p =* 0.83).

### 3.1. Demographic Information in Positive Patients

Although 22 patients had at least one positive sample (22.9% of the 96 participants), positive samples collected from the antrum and fundus also were counted separately yielding 26 positive samples (13.5% of all biopsy samples analyzed). More positive samples were from the fundus (53.9%, *n* = 14) compared with the antrum (46.2%, *n* = 12). Both the antrum and fundus biopsies were positive in only 4 patients (18.2% participants (Table 2). Normal gastric findings were uncommon among those with positive biopsy samples (2 of 22 patients; 4.5%) compared with 12 of the 74 negative patients (16.2%). Gastritis was the most common gastric finding, in 62.5% of all participants, 59.0% of those with a positive biopsy and 63.5% of those with a negative biopsy (*p* = 0.8). Regardless of biopsy outcome, epigastric pain was the most common reason listed for undergoing the EGD (17.7%), followed by abdominal pain (13.5%) (Table 2).

### 3.2. PCR Amplification of cagA Gene Variable 3′ Region and EPIYA Motif Bioinformatic Analysis

Detection for 3′ variable region of the *cagA* gene in the 26 DNA samples from *H. pylori* positive strains resulted in 20 positive samples that were confirmed by agarose gel electrophoresis, indicating *cagA+* infection in those patients (57.6%; Table 3). Isolates that did not generate amplification of *cagA* 3′ DNA end, were considered *cagA* negative and excluded from further molecular analysis. All *cagA* sequences from these isolates aligned well with the *cagA* 3′ ends of reference strains 26695 and 60190. The most common EPIYA pattern was the ABC (11/20 = 55%), ABCC (3/20 = 15%), with one AB pattern (5%) (Table 3).

Based on CLUSTALW and Mega 7.0 sequence alignment, all sequenced *cagA* variable region showed the following patterns, EPIYA(K/Q)VNKKK(A/T)GQ that corresponds to EPIYA-A; the pattern E(P/S)IY(A/T)(Q/K)VAKKV(N/T)(A/Q)KI, to EPIYA-B; and the pattern EPIYATIDDL(G/R) to EPIYA-C (Figure 1).

Within this samples, no isolates corresponded to the EPIYA-D Eastern type: EPIYATIDFDEANQAG. The relationship between the number of EPIYA-C motifs or *cagA* types and gastric disease was not as clear because of the low number of isolates with more than one EPIYA C domain. Table 3 shows that as the number of EPIYA C motifs increased, the frequency of gastric disease decreased. *H. pylori* strains with only one EPIYA-C motif (ABC pattern) in the *cagA* proteins were more frequently associated with atrophic gastritis. Strains with two EPIYA-C motifs were more common among participants with intestinal metaplasia and erosive gastropathy. There was an almost equal distribution in the EPIYA B motif, with the threonine (53.3%) substituted for alanine (46.5%) as EPIYT instead of EPIYA (Figure 1). Phylogenetic analysis of *cagA* positive biopsies resulted in four distinctive groups. Group A contained eight of positive isolates which also contained the reference strain 26695, the Colombian isolate co5017 and the Mexican 373H isolate. Group B had five isolates which also contained the Mexican isolate 10N. In group C, two isolates were found which grouped closely to a Colombian co5007 and Asian isolate, F32 (Appendix A).

### 3.3. Presence of vacA in Native American Isolates

All analyzed isolates expressed the *vacA* gene. Two biopsies provided a mixed infection with s1a,s1b. The combination of s1b (70%) and m1 (75%) alleles was the most frequent; while allele s1a was found in three samples with m2 found in five samples (Table 3). Most of the isolates identified located with reference strains USA2754 and J99 indicating expression of the *vacA* s1b. One *vacA* s2 isolate placed with reference strainTx30a; while three others were s1a and located with reference strains 26695 and AR-710 (Figure 2). When analyzing the *vacA*m allele, two isolates with a *vacA*m1 allele (6F and 38A) were found closely associated with the m2 group (Appendix A). One *vacA*m2 isolate (31A) grouped closely to Colombian (NA1692) and Alaska (18 and 7), typical Amerindian isolates. Fewer *vacA*m2 genotypes were found in *H. pylori* in this study. The *cagA^+^ vacA* s1bm1 genotype was the most frequent isolate identified and patients with gastritis Table 3).

## 4. Discussion

*H. pylori* is recognized as a major causative agent of chronic gastritis and peptic ulcer disease and has been identified as a major risk factor for development of gastric cancer. Native Americans in the southwestern region of the United States are at increased risk for development of gastric cancer. While studies looking at the association of *H. pylori* and gastric pathologies have taken place in Indigenous populations in Alaska and Canada [46,48,57,58,59], this is the first attempt to characterize the *H. pylori* genotypes and their association with gastric diseases among Navajo adults in northern Arizona.

*H. pylori* infection in Native American and Indigenous communities in Alaska is usually higher than their non-native counterparts and is comparable to that of developing countries, which may help explain the high rate of gastric cancer [40,41,42,43,44]. In a random sample of households of three communities on the Navajo Nation, we reported 56.4% prevalence and 72% of households had at least one person with a positive Urea Breath Test [5]. A retrospective study among Navajos living in New Mexico, reported seroprevalence for *H. pylori* of 74.4%; furthermore, 69.23% of patients with intestinal metaplasia were also positive for *H. pylori* [40]. Among Alaska Natives, 65% prevalence was reported for *H. pylori* on histology, and prevalence increased in patients with gastritis (66%) and chronic gastritis (87%) [44]. Prevalence using serologies is higher in some studies, with infection prevalence among Alaska Natives near 75% [44,46].

Gastric cancer is a multifaceted disease, with a wide range of associated risk factors, such as diet, obesity and chronic health conditions, and access to care and health care utilization [59,60,61]. However, because of the high prevalence of *H. pylori* and the disproportionately high incidence of gastric cancer, it is likely that *H. pylori* plays a central role in the gastric cancer observed among Native Americans and Alaska Natives [53]. Alaska Natives experience 4 times the mortality due to gastric cancer when compared to non-Hispanic White [62]. It is unknown whether genotypic differences in the circulating pathogen strain might also explain the high morbidity and mortality.

In our study, we collected gastric biopsies from 96 Navajo patients undergoing a scheduled EGD procedure. When biopsies were analyzed for presence of *H. pylori*, 22 out of 96 patients were positive (22.9%). This value is much lower than the near 60% observed using the urea breath test in these communities, most likely because it includes all biopsies, including those which may have been taken to confirm post-treatment *H. pylori* eradication. It is likely that many of the processed biopsies originated from patients that had or were currently undergoing antibiotic therapy. Because we were interested in the virulence factors, we did not subset the data. The 22.9% prevalence in EGD samples should not be used as an estimate of *H. pylori* prevalence in this study population.

The clinical relevance and geographical distribution of the virulent genotypes of *H. pylori* is still a matter of debate. Here, we report on the prevalence and relationship of virulence genes of *H. pylori* (*vacA* and *cagA*) with clinical status in patients from the Navajo Nation. The presence of *cagA* has been commonly associated with an increased risk for the development of gastric cancer [22,63]. In East Asia more than 90% of strains possess *cagA* which may help explain the high prevalence of gastric cancer in Asian countries [26,64]. We observed a 77% prevalence of *cagA* positive *H. pylori* in our isolates which is close to the 74% reported prevalence in Mexican patients with different types of chronic gastritis [50] and the 83.8% in Colombian patients with diverse gastric pathologies [37,65].

The type and number of EPIYA motifs in the 3′ region of *cagA* are predictors of virulence because of enhanced *cagA* phosphorylation [28,32,65]. Despite the limited number of patients with gastric atrophy and intestinal metaplasia in this series of patients, our data trend toward increasing number of EPIYA-C motifs being associated more severe disease (Table 3). Indeed, studies using *H. pylori* strains isolated from Japanese participants have shown that strains expressing *cagA* with higher numbers of EPIYA-C motifs were more prevalent in patients with atrophic gastritis and gastric cancer [33,66]. Another study in South Africa found that *H. pylori* strains expressing *cagA* proteins with four or more variable-region EPIYA-C motifs originated from patients with gastric cancer [51]. Further, Colombian studies showed not only these strains were associated with more severe lesions, but they were also more frequently found [30,37,65].

While we were unable to identify isolates with more than two EPIYA-C domains in this study population; 34.3% to 54.5% were reported in Colombian [22,60], 30% in Mexican [67], and 11.5% in Peruvian isolates [68]. Our study, as well as in a Mexican study investigating virulence factors in different Indigenous groups in Mexico, found that the most common *vacA* EPIYA domain was ABC [69]. None of the *H. pylori* strains in our study contained an EPIYA-D motif.

We found a predominance of *cagA*+ *vacA*s1m1 genotypes (76.9%) in the *H. pylori* isolates. Although all *H. pylori* strains express the *vacA* gene, its expression and structure may vary resulting in polymorphisms inducing different levels of cytotoxicity and gastric pathologies [70,71]. These polymorphisms are in the 5′ end, in the signal (s) and medial (m) regions for which strains expressing the combination of alleles s1m1 are capable of more cytotoxic activity in vitro than those expressing the *vacA*s2m2 alleles [33]. Our results are consistent with research in Latin America [49] and indigenous communities [26]. In a high-risk region from northeastern Brazil, a predominance of *cagA*+ *vacA*s1m1 isolates was reported, which were associated with more severe gastric pathologies [31]. In Mexico, the *vacA* s1m1 genotype and *cagA* EPIYA-ABC motif were present in patients with chronic gastritis [50]. The same pattern was observed in Peruvian Amerindians [71] and in Indigenous communities in Mexico [68].

The limited association between disease and virulent genotypes of *H. pylori* has provided conflicting results. In Alaska Natives 56% of isolates expressed *vacA*s1 and 39% s2 subtype where the genotype *cagA*+, *vacA*s1m1 was associated with a more severe pathology [45,46,47,48]. Furthermore, the *vacA* s1m1 genotype was the most prevalent. Our data are consistent with Miernyk et al. [48], although the infection prevalence among biopsies was low (22.9%), the prevalence of *cagA* positive *H. pylori* was 77%, *vacA* s1m1 was 80.7%. Further, western *cagA* positive strains with EPIYA ABC motifs were always associated with gastritis and a high percentage of *vacA* s1bm1 genotypes.

Although this is the first report on the expression of virulence genes in *H. pylori* isolates from the Navajo Nation, this study was limited by sample size. We tested biopsy samples from 96 participants with only 26 samples testing positive for *H pylori.* When comparing *cagA* and *vacA* to the participant’s histopathology findings, we did not have a large enough sample to definitively assess the association. Also, other microbial virulence factors (*homB, oipA, babA, dupA*) must also be considered [13,16]. We are continuing to recruit participants in this population and these issues will be addressed.

While our project focused on a relatively limited geographic area, this is an area where *H. pylori* prevalence and gastric cancer rates are disproportionally high, and implications of virulence factors is yet unknown. Gastric cancer is a complex disease with multiple etiologies each with a differing suite of risk factors [60]. Environmental factors have been shown to play an important role and will need to be considered [5,61]. While it is decreasing in most of the United States, a recent study showed that rates are almost universally higher among American Indians/Alaska Natives compared with geographically proximal White populations [58]. Improved understanding of the etiology of gastric cancer in the Navajo Nation will allow for incorporation of identified risk factors to improve prognosis.

## Figures and Tables

**Figure 1 diseases-10-00019-f001:**
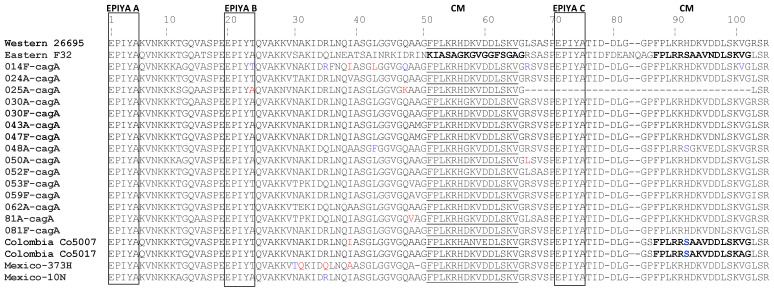
Amino acid sequence of the CagA 3’ terminal region from strains containing three EPIYA motifs. The reference strain F32 was isolated from a Japanese patient and it is representative of an Eastern sequence (ESS; AAF17597.1). The reference strain 26695 (NP207343.1) is representative of a Western CagA-specific sequence (WSS). The strains Co5007 (EU251000.1), Co5017 (EU250993.1) and 373H (JN390445.1) and 10N (JN390446.1) were isolated from indigenous groups in Colombia (Co) and Mexico, respectively. Boxes illustrate the regions containing the EPIYA domains and underline shows the *cagA* multimerization motif (CM).

**Figure 2 diseases-10-00019-f002:**
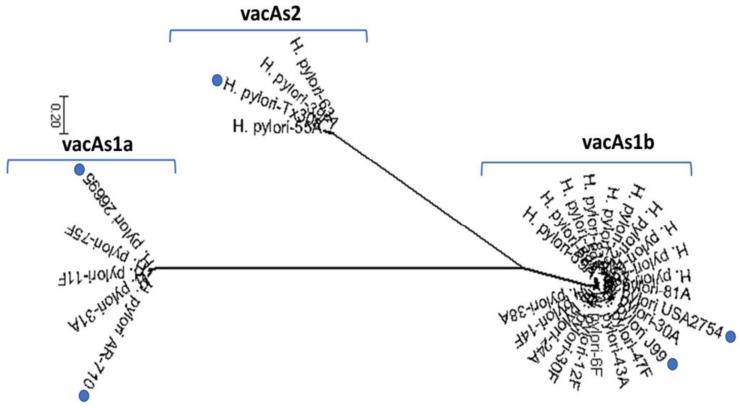
Phylogenetic tree of Navajo *vacA*s alleles in Helicobacter pylori sequences. The *vacA*s1a alleles are shown on the left with two typical *vacA*s1a strains (26695-NC_000915.1:938415-942287 and AR-710-AY185128.1). Sequences shown on the right display *vacA*s1b alleles with reference strains (USA2754-AB057223.1, 60190-U05676.1 and J99-NC 000921.1). The *vacA*s2 allele is shown on the top with a representative strain (Tx30a-U29401.1) in this group.

**Table 1 diseases-10-00019-t001:** Primers used for amplification and sequencing of 16S rRNA, *cagA*, and *vacA* genes.

Primer	Use	Sequence	Target	Reference
16S1/16S2	Amplification	5′ GCTAAGAGATCAGCCTATGTCC 3′ 5′ TGGCAATCAGCGTCAGGTAATG 3′	*16S rRNA*	[51]
VA1-F/VA1-R	Amplification and sequencing	5′ ATGGAAATACAACAAACACAC 3′ 5′ CTGCTTGAATGCGCCAAAC 3′	*vacA* s	[33,50]
mF1/mR1	Amplification and sequencing	5′ ACCGCTCATBAAGATYAAYARCGCTC 3′ 5′ GCTAGGCGCTCTTTGAATTGC 3′	*vacA* m	[52]
glmM-F/glmM-R	Amplification	5′ TAACCGAAGACATGCGCTG 3′ 5′ CATGAAAGATTTCTTCAATCAATCGCT 3′	*glmM*	[52]
*cagA*-F1/ *cagA*-R1	Amplification	5′ GATAACAGGCAAGCTTTTTGAGG 3′ 5′ CTGCAAAAGAATGTTTGGCAG 3′	5′ *cagA*	[53]
CAGTF/ CAGTR	Amplification	5′ ACCCTAGTCGGTAATGGGTTA 3 5′ GTAATTGTCTAGTTTCGC 3′	3′ *cagA*	[53]
*cagA*-2/ *cagA*-4	Sequencing	5′ GGAACCCTAGTCGGTAATG 3′ 5′ ATCTTTGAGCTTGTCTATCG 3′	3′ *cagA*	[54,55]

**Table 2 diseases-10-00019-t002:** Comparing demographic and clinical characteristics by positive or negative biopsies.

	Total	Positive	Negative	*p* Value
Total	96	22	74	
Age (mean)	55.6	54.0	56.1	0.57 *
Gender				0.08 #
Male	25	10	15	
Female	71	12	59	
Reason for Endoscopy				
Epigastric Pain	17	5	12	0.153 §
Abdominal Pain	13	4	9	
GERD	10	1	9	
Bloating	10	2	8	
Right Upper Quadrant Pain	6	2	4	
Nausea	5	1	4	
Left Upper Quadrant Pain	4	1	3	
Gastritis	3	1	2	
GI Bleed	2	2	0	
Constipation	2	0	2	
Gastric Finding				
Normal	14	2	12	0.21 §
Gastritis	60	13	47	
Gastric Atrophy & Fundal Gastritis	4	3	1	
Erosive Gastropathy	4	3	1	
Gastritis & Intestinal Metaplasia w/o Dysplasia	7	2	5	

* *t* test; # X^2^ square; § Fisher’s exact test.

**Table 3 diseases-10-00019-t003:** Gastric findings and scores in Navajo patients and their association with *H. pylori vacA* and *cagA* genotypes.

	*cagA*	*vacA* Genotype					
Gastric Disease	EPIYA	s1a,s1b	s1a	s1b	s2	m1	m2
**Normal**							
		0	0	0	1	0	2
**Gastritis**							
	AB (1)	0	0	1	1	1	1
	ABC (7)	1	1	9	1	8	2
**Gastritis & ulcer**							
**Erosive gastropathy**							
	ABC (2)	1	1	1	0	2	0
**Gastric atrophy & Fundal gastritis**							
	ABC (2)	0		1	0	1	0
	ABCC (1)	0	1	0	0	1	0
**Gastric & Intestinal Metaplasia**							
	ABCC (1)	0	0	1	0	1	0
**Gastritis & Intestinal metaplasia without dysplasia**							
	ABCC (1)	0	0	1	0	1	0
**TOTAL**	**(15/20)**	**2**	**3**	**14**	**3**	**15**	**5**

## Data Availability

These sequence data have been submitted to GenBank database under accession number MT875170, MT878479-MT878479. GenBank: www.ncbi.nlm.nih.gov (accessed on 15 June 2021).

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
