# Peer review of "Helicobacter pylori in Native Americans in Northern Arizonaâ€"

_diseases, 2022, doi:10.3390/diseases10020019_

Round 1
Reviewer 1 Report
The original article entitled “Helicobacter pylori in Native Americas in Northern Arizona” contains properly conducted research, the description of which is clear and understandable. Below I would like to present a list of minor suggestions that need to be considered:
- “... gram-negative bacillus” -> Gram-negative rod (page 1)
- CagA or VacA protein (should be capitalized when discussing proteins; please change in all places needed, eg. Page 2)
- Point 2.4. “16S rDNA gene” -> 16S rRNA gene
- please add a space to the capitation of the Table 2 (the text merged)
- In the description of the results under the Table 2, it seems to me that the statistical significance of the differences between men and women is 0.08 and not as described 0.22
- Please correct the font size of the text on page 7
Recommendation for the future:
Although it was mentioned in the discussion in the limitations, I believe that it is definitely worth increasing the pool of tested strains and extending the research to other virulence genes in the future. Then the context of the results obtained will be more important.
Author Response
The original article entitled “Helicobacter pylori in Native Americas in Northern Arizona” contains properly conducted research, the description of which is clear and understandable. Below I would like to present a list of minor suggestions that need to be considered:
- Minor suggestions have been addressed in the text of the manuscript.
- In the description of the results under the Table 2, it seems to me that the statistical significance of the differences between men and women is 0.08 and not as described 0.22. That is correct and it is now in the table.
- Please correct the font size of the text on page 7
Recommendation for the future:
Although it was mentioned in the discussion in the limitations, I believe that it is definitely worth increasing the pool of tested strains and extending the research to other virulence genes in the future. Then the context of the results obtained will be more important.
I agree 100% with reviewer 1. Unfortunately, getting approval of this publication by the Navajo Nation IRB took almost 1.6 years! This was a Pilot project that led to funding of a full project to look at different parts on the Navajo Nation. That is the reason why this manuscript was a bit short. Nevertheless, a seminal paper with 350 sets of biopsies will become available for publication later this year. We are currently examining virulence factors in these samples which will serve to come full circle. Thank you for you timely suggestion.
Reviewer 2 Report
This manuscript by Fernando P. Monroy et al. describe interesting genetic study of Hp strains infecting Native Americans.
This manuscript deserve extensive modification before possible consideration for publication.
Global :
Number below twelve have to be written in full letters.
Introduction :
Intra-gastric repartition of Hp strains has to be introduced completely. See for example Pichon et al. J. Clin. Med. 2020, 9(9), 2812; https://doi.org/10.3390/jcm9092812
Methods :
Manufacturer's informations haev to be correctly reference, when they first (and only when they first) appear in the manuscript)
Why have the authors not considered to analyse other virulence factors? Why did they wait for other inclusion ?
Results :
Figure 1 and following paragraph have not the same police as the whole manuscript.
The authors described mixed infection. Did theyr explore respective proportions ot the different subpopulations?
Author Response
Introduction :
Intra-gastric repartition of Hp strains has to be introduced completely. See for example Pichon et al. J. Clin. Med. 2020, 9(9), 2812; https://doi.org/10.3390/jcm9092812
Thank you for providing this reference, it is of great interest and beyond the scope of this research. While Pichon et al. had access to three patients undergoing sleeve gastrectomy to determine gastric distribution and density of H. pylori in the different gastric regions, we are limited to two biopsies as authorized by medical procedures in place at Indian Health Service Clinics. Getting approval for two biopsies was very difficult and we must respect policies that are in place to safeguard Native American genetic integrity.
Why have the authors not considered to analyse other virulence factors? Why did they wait for other inclusion ?
We are currently examining virulence factors in the latest cohort of gastric samples. It was not possible for us to go back and analyze virulence factors in the samples in this manuscript as it would have needed Navajo Nation IRB approval for this type of study and having to re-consent these individuals who are difficult to locate because of their mobility within the reservation. Our new consent form authorizes the study of virulence factors in current and future biopsies.
The authors described mixed infection. Did theyr explore respective proportions ot the different subpopulations?
No, we found two patients with mix vacA s1/s2 and were left out of the study. Mixed infections are very common in the literature and can range from 2% to 12% of positive H. pylori positive samples. Pichon et al., also describe mixed infection in their manuscript.
Round 2
Reviewer 2 Report
The Authors have answered all my previous comments thanks.
Author Response
thank you!